# Measuring the Contribution of Leaves to the Structural Complexity of Urban Tree Crowns with Terrestrial Laser Scanning

**Georgios Arseniou** [1], **David W. MacFarlane** [1,*] and **Dominik Seidel** [2]

1   Department of Forestry, Michigan State University, East Lansing, MI 48840, USA; arseniou@msu.edu
2   Department of Silviculture and Forest Ecology of the Temperate Zones, University of Göttingen, 37077 Göttingen, Germany; dseidel@gwdg.de
*   Correspondence: macfar24@msu.edu

**Abstract:** Trees have a fractal-like branching architecture that determines their structural complexity. We used terrestrial laser scanning technology to study the role of foliage in the structural complexity of urban trees. Forty-five trees of three deciduous species, *Gleditsia triacanthos*, *Quercus macrocarpa*, *Metasequoia glyptostroboides,* were sampled on the Michigan State University campus. We studied their structural complexity by calculating the box-dimension ($D_b$) metric from point clouds generated for the trees using terrestrial laser scanning, during the leaf-on and -off conditions. Furthermore, we artificially defoliated the leaf-on point clouds by applying an algorithm that separates the foliage from the woody material of the trees, and then recalculated the $D_b$ metric. The $D_b$ of the leaf-on tree point clouds was significantly greater than the $D_b$ of the leaf-off point clouds across all species. Additionally, the leaf removal algorithm introduced bias to the estimation of the leaf-removed $D_b$ of the *G. triacanthos* and *M. glyptostroboides* trees. The index capturing the contribution of leaves to the structural complexity of the study trees (the ratio of the $D_b$ of the leaf-on point clouds divided by the $D_b$ of the leaf-off point clouds minus one), was negatively correlated with branch surface area and different metrics of the length of paths through the branch network of the trees, indicating that the contribution of leaves decreases as branch network complexity increases. Underestimation of the $D_b$ of the *G. triacanthos* trees, after the artificial leaf removal, was related to maximum branch order. These results enhance our understanding of tree structural complexity by disentangling the contribution of leaves from that of the woody structures. The study also highlighted important methodological considerations for studying tree structure, with and without leaves, from laser-derived point clouds.

**Keywords:** terrestrial laser scanning; fractal dimension; box-dimension; foliage; urban ecology; *Gleditsia triacanthos*; *Quercus macrocarpa*; *Metasequoia glyptostroboides*

## 1. Introduction

Trees have an inherent fractal-like branching architecture [1–3] mirroring principles of fractal geometry [4]. However, tree branching networks are not perfect fractals, lacking self-similarity across all scales of the branching hierarchy [1,5,6]. Nonetheless, major theories linking tree morphology to tree physiology (e.g., pipe model theory [7–10]; metabolic scaling theory [11]) and mechanical stability (e.g., resisting wind stress [12]) have been advanced by assuming that the fractal-like character of trees explains the structural complexity of their crowns [13] and how they grow to occupy space [14,15]. One of the main challenges in testing such theories is finding reliable ways to accurately measure the structural complexity of trees in a way that reflects the fractal dimension of tree crowns.

The growing environment of a tree affects its crown architecture and competition for light from neighboring trees [16] significantly disrupts the inherent fractal-like character of trees growing in forest stands and plantations [17,18]. According to Seidel [13], Douglas-fir trees growing in forest gaps had more complex crowns compared to trees of the same

species growing in closed canopy conditions; this implies that the light regime significantly affects the fractal dimension of a tree, which negatively relates to competition [19]. Therefore, we expect that the typically lower number, or complete absence of, neighboring trees in cities should allow urban trees to better express their inherent fractal character; this was an important reason to focus on urban open-grown trees in this study. Of course, cities have heterogeneous growing conditions [20–26], characterized by anthropogenic barriers to root and crown expansion [27–29], systematic tree pruning [29,30], increased atmospheric temperatures and reduced water infiltration [22,31,32], air pollutants [33], and heterogeneous soil properties [34,35], which can affect the fractal dimension of tree crowns [36]. Nonetheless, the inherent fractal-like character of open-grown trees should be more evident compared to trees growing in competition with other trees.

Open-grown trees can be found both in urban and rural forest conditions, but for urban conditions there is a shortage of robust models. This limits our understanding of the basic ecological services of urban forests [37], despite the fact that urban trees provide a range of significant ecological services, e.g., carbon storage [35,38–40], air pollutant uptake [41–43], water purification, pollination, biodiversity, and energy savings for buildings [23,24,41,44]. In order to optimize the benefits of urban forests, we need to study the structure and function of trees in cities. For example, we know that the fractal dimension of tree crowns relates to their ability to tolerate shade [45,46], which affects the shading benefits of trees, as well as their ability to tolerate the drought and the heat of cities [36].

New advances in terrestrial laser scanning (TLS) technology allow for accurate, direct measurements of the three-dimensional structure of trees [6,47] and many studies have utilized TLS to quantify stem profiles and timber volume [48–52], leaf and crown attributes [53–55], and above-ground tree biomass [56–61]. TLS creates 'point clouds' of trees by emitting laser pulses and analyzing the returned energy as a function of either time (time-of-flight systems) or shift in the phase of the light wave of the emitted laser beam (phase-shift technology) [56,62]. One way to generate data for analyzing the fractal-like character of tree branching networks from TLS point clouds is the generation of Quantitative Structure Models (QSMs), by fitting cylinders to a tree's point cloud that preserve branch and stem topology [63–67]. Lau et al. [68] generated QSMs of tropical trees to study the theoretical scaling exponents derived from the metabolic scaling theory [11] that describes the fractal-like structure of trees.

Another approach is the "box-counting" method [15], which considers the number of boxes that are needed to encapsulate all points of a laser-scanned tree, as box size iteratively reduces. Seidel [13] showed how the "box-dimension" metric can be calculated from the point cloud of a tree to describe its fractal dimension in terms of structural complexity. The box-dimension metric has no units and its possible values range between one and three. Trees with great structural complexity and "space-filling character" have box-dimension values closer to three, whereas a box-dimension value equal to one implies a perfectly cylindrical stem with no branches, e.g., a dead tree [13]. Box-dimension values smaller than one imply that the lower "cut-off" has not been properly defined because the mean distance between points is greater than the edge-length of the smallest box. Values of three (or greater) would imply that a tree is a solid cube, which is not valid. The box-dimension is a more direct and simple way to measure the fractal-like character of a tree because it lacks the assumptions and stochasticity inherent in QSMs, using only the raw point cloud data generated by TLS.

Leaves increase uncertainty in the underlying branching architecture because they occlude underlying branches and move more in the wind [47,69]. Davison et al. [70], for example, showed how leaf phenology affects the estimation uncertainty of metrics of forest structural diversity when laser scanning data are used. "Leaf-off" laser scanning data can provide better estimates of the crown architecture of deciduous tree species [70] because leaf occlusion effects are avoided.

There are several studies that have explored how leaf-off and leaf-on airborne laser scanning data compare for the estimation of forest volume and other forest inventory

attributes [71–74], but few, if any, have examined the effects of leaf-on computation of the fractal metrics of tree branching architecture. Perhaps more importantly, we lack a basic understanding regarding the role of foliage in the structural complexity of trees, which is fundamental to understanding how trees position their leaves and branches to maximize light capture and minimize self-shading [45,46,75], optimize crown architecture to improve water transport and resist drought [36], and reduce wind stress [12,76,77], which has been shown to be affected by both the increased drag of foliage [78,79] and the uncertain effects of branches.

Artificial leaf separation from the leaf-on point clouds of trees is a promising methodology to deal with the fact that trees can't always be scanned in a leaf-off condition (e.g., evergreen trees). There are three main types of algorithms to separate the leaf from the woody material of laser point clouds of trees: (1) algorithms that use the geometry of laser points, (2) algorithms that consider the radiometric properties of the returned laser pulses, and (3) algorithms that combine the previous approaches [80–83]. The radiometric-based algorithms assume that the leaves and the woody material of trees have different intensity characteristics at the wavelength of the laser scanner, which depend on the laser scanning distance, the incidence angle, and the technical characteristics of each instrument [83]. However, the geometry-based algorithms consider only the 3D coordinates of the points of a laser-scanned tree based on supervised machine learning [80,84] or unsupervised classification methods [81,83]. In general, we still need a better understanding of the effect of these classification algorithms for leaf separation when studying tree architecture [81].

In this study, we used the box-dimension metric to quantify the structural complexity of three deciduous tree species in their leaf-on and leaf-off conditions. Furthermore, we artificially removed the leaves from the tree point clouds generated from leaf-on data, using the *TLSeparation* algorithm [85], and we computed the box-dimension metric for the leaf-removed tree point clouds. The questions that we wanted to answer were the following:

- How do changes in leaf condition of deciduous tree species with different leaf types affect their structural complexity?
- How do differences in the contribution of leaves to the structural complexity of the study trees relate to their above-ground architecture?
- What is the effect of artificial leaf removal from leaf-on tree point clouds on estimated structural complexity? Is there an error in estimating the structural complexity of the tree point clouds after artificial leaf-removal, compared to leaf-off point clouds of the same trees?
- How does potential error in estimating the structural complexity of the tree point clouds due to the artificial leaf removal relate to the branch architecture of trees?

We hypothesized that the leaves of trees would significantly increase their structural complexity because the irregular outline shape of leaves is fractal-like [86–88], and the presence of foliage implies that more space is occupied by a tree and, consequently, more laser points are captured in its crown. So, a larger number of boxes is required to encapsulate all points of the laser-scanned tree, which results in a greater value of the box-dimension metric [13,89,90]. Furthermore, we hypothesized that differences in the contribution of leaves to tree structural complexity have ecological importance because differences should relate to self-shading of tree crowns [91], the shade tolerance of the tree species, and the type and shape of the leaves [36,92].

We also hypothesized that errors in estimation of the box-dimension resulting from artificial leaf removal, would relate to the type of leaf (broad vs. needle and compound vs. simple) and the order and the size of the branches of a tree, the latter of which because the point cloud density can change across the branching network of a tree and leaf separation algorithms are sensitive to this [80,81].

## 2. Materials and Methods

### 2.1. Urban Tree Data

Forty-five trees of three species, representing different tree functional types, were sampled on the Michigan State University campus: sixteen *Gleditsia triacanthos* (honey locust) trees, which are compound-leaved, deciduous angiosperms, fifteen *Quercus macrocarpa* (bur oak) trees, which are entire-leaved, deciduous angiosperms, and fourteen *Metasequoia glyptostroboides* (dawn redwood) trees, which are needle-leaved, deciduous gymnosperms (Figure 1). The trees were selected to cover a large range of sizes within each species (see Table 1).

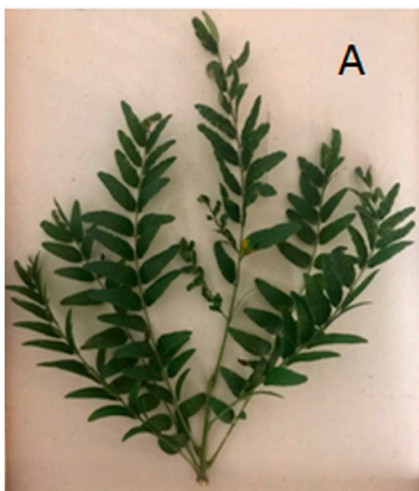 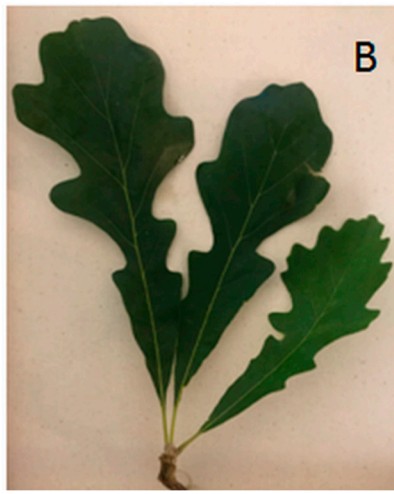 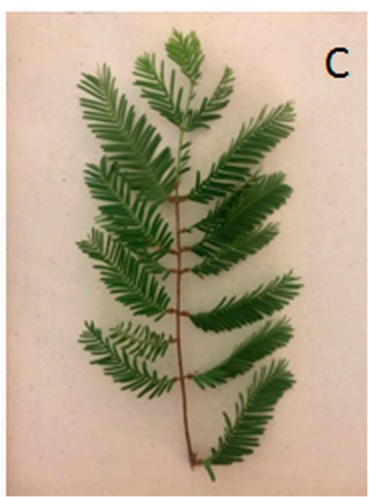

**Figure 1.** Sample of leaves of the species (**A**) *G. triacanthos* (**B**) *Q. macrocarpa* (**C**) *M. glyptostroboides*.

**Table 1.** Summary statistics resulting from different measurements of tree size and structural complexity.

| Summary Statistics | All Trees | *Gleditsia triacanthos* | *Quercus macrocarpa* | *Metasequoia glyptostroboides* |
|---|---|---|---|---|
| no. trees | 45 | 16 | 15 | 14 |
| DBH (cm) * (mean [min, max]) | 54.1 [15, 122.2] | 52.9 [18.4, 72.8] | 58.8 [29, 83.8] | 50.5 [15, 122.2] |
| Height (m) (mean [min, max]) | 13.8 [4.4, 24.1] | 12.5 [10.4, 18.4] | 15.8 [9.1, 21.3] | 13.1 [4.4, 24.1] |
| WSA (m$^2$) ** (mean [min, max]) | 204.2 [29.9, 467.0] | 265.4 [65.2, 408.6] | 225.4 [60.4, 467.0] | 111.5 [29.9, 250.2] |
| Stem WSA (m$^2$) (mean [min, max]) | 13 [2.1, 44.6] | 11.4 [4.1, 20.1] | 16.2 [4.7, 30.3] | 11.4 [2.1, 44.6] |
| Branch WSA (m$^2$) (mean [min, max]) | 191.2 [27.7, 436.7] | 253.9 [61.2, 395.5] | 209.2 [55.7, 436.7] | 100.1 [27.7, 231.8] |
| $D_b$-leaf.on (mean [min, max]) | 2.06 [1.89, 2.23] | 2.09 [1.89, 2.20] | 2.03 [1.91, 2.11] | 2.07 [1.94, 2.23] |
| $D_b$-leaf.off (mean [min, max]) | 1.97 [1.82, 2.11] | 2.02 [1.84, 2.11] | 1.92 [1.82, 2.04] | 1.97 [1.84, 2.1] |
| $D_b$-leaf.rm (mean [min, max]) | 1.9 [1.76, 2.14] | 1.84 [1.76, 2.0] | 1.93 [1.83, 2.03] | 1.93 [1.8, 2.14] |
| LCC index (mean [min, max]) | 0.04633 [0.00064, 0.16394] | 0.03273 [0.01371, 0.0762] | 0.05867 [0.00667, 0.10883] | 0.04864 [0.00064, 0.16394] |
| %RE (mean [min, max]) | 5.55 [0.17, 14.64] | 8.91 [1.07, 14.64] | 2.43 [0.17, 5.46] | 5.06 [0.92, 11.53] |
| Mean Path length (m) (mean [min, max]) | 12.9 [3.7, 23.9] | 14.8 [9.5, 22.0] | 14 [6.9, 23.9] | 9.5 [3.7, 18.6] |
| Max. Path length (m) (mean [min, max]) | 22.8 [6.5, 42.7] | 24.8 [17.3, 37.5] | 24.9 [12.3, 42.7] | 18.3 [6.5, 35.8] |
| 25th % Path length (mean [min, max]) | 10.9 [3, 20.6] | 13.2 [7.7, 18.1] | 11.7 [5.4, 20.6] | 7.4 [3, 14.9] |
| # of branch orders (median [min, max]) | 5 [1, 11] | 5 [1, 11] | 5 [1, 10] | 4 [1, 9] |

* DBH = Diameter at Breast Height, ** WSA = Woody Surface Area.

The *G. triacanthos* and *Q. macrocarpa* trees were laser-scanned with leaves-on in July and August 2019, and the *M. glyptostroboides* trees were laser-scanned with leaves-on in August 2020 (see specific methods below). The same trees were also laser-scanned in leaves-off condition between January and March 2020. Before re-scanning the study trees, we confirmed that none of them were pruned between the leaves-on and leaves-off scans by the Michigan State University arborists. Therefore, pruning did not cause any bias in the quantification of the structural complexity of the trees during the study period. Following this experimental design, any change in the structural complexity of the study urban trees between the leaves-on and leaves-off scans should be attributed only to changes in their foliage, not their branching architecture. Of course, tree pruning prior to the study should have an effect on the crown architecture of the study trees, but it did not influence the changes in their structural complexity during the study period.

### 2.2. Terrestrial Laser Scanning and Point Cloud Processing

The FARO Focus$^{3D}$ × 330 terrestrial laser scanner was used to scan the trees. This laser scanner operates with laser light of 1550 nm wavelength, typical beam divergence 0.19 mrad, and a range of 0.6–330 m. In order to minimize occlusion effects in the point clouds, each individual tree was scanned at high resolution from a minimum of four different directions at different distances, and five reference target-spheres were placed around a laser-scanned tree to spatially reference all scans and create a single point cloud for each tree, following the field scanning protocols suggested by Wilkes et al. [69]. The first two scans were conducted in opposite directions, from distances that allowed the top of the focal tree to be clearly visible. The other two scans were also conducted in opposite directions (perpendicularly to the first two scans) but from a closer distance to the tree, to better capture its branching architecture and get closer views of the main stem. Two or three additional scans were conducted underneath the crown of large trees with wide crowns in order to capture more dense point clouds of the branches. All laser scans were conducted when there was little or no wind.

The software SCENE 2019.2 (FARO Technologies Inc., Lake Mary, FL, USA, 2019.2) was used to spatially co-register and noise-filter all scans in an automatic way. With the same software, each tree was then manually separated from the point cloud of the urban site background. This process has been shown to be an accurate alternative to a fully automatic segmentation process [93].

### 2.3. Leaf and Wood Classification of the Point Clouds

The *TLSeparation* algorithm [85] was applied to the point clouds of the trees with their leaves-on. This algorithm separates points that belong to the woody components of the trees from points that belong to their foliage, based on unsupervised classification of geometric features (leaf and wood materials within the point cloud have different spatial arrangement) and "shortest-path" analysis, which facilitates detection of paths through the branching network (from tree base to branch tip) with high occurrence frequency [81]. This approach was used to generate a single point cloud for each tree containing only points classified as woody parts of the tree.

### 2.4. Quantification of the Structural Complexity of Trees

The box-dimension metric ($D_b$), which is derived from fractal geometry principles [4], was used to quantify the above-ground structural complexity (fractal dimension) of the trees [90] in three conditions: (1) leaf-on, (2) leaf-off, and (3) after the leaves were artificially removed from the leaf-on point clouds. The box-dimension equals the slope of the least-squares line when the logarithm of the number of boxes required to capture all points of a laser-scanned tree is regressed against the logarithm of the inverse of the size of a box relative to the size of the initial box, which is the smallest box encapsulating the whole tree, i.e., "upper cut-off" (Figure 2, [13,90]). The intercept of the regression line describes the size of the crown of a tree (i.e., crown radius, [19]). The size of the smallest box ("lower cut-off")

was 10 cm in this study. This was selected based on a very liberal estimate of the maximum distance between two neighboring laser points at any given location in the tree because the "lower cut-off" must ensure that no box is empty due to missing data, i.e., it fits in the "unsampled" space of a scanned tree. The algorithm written in Mathematica 12.2 [94] for the computation of the $D_b$ metric is available online as Supplementary Materials.

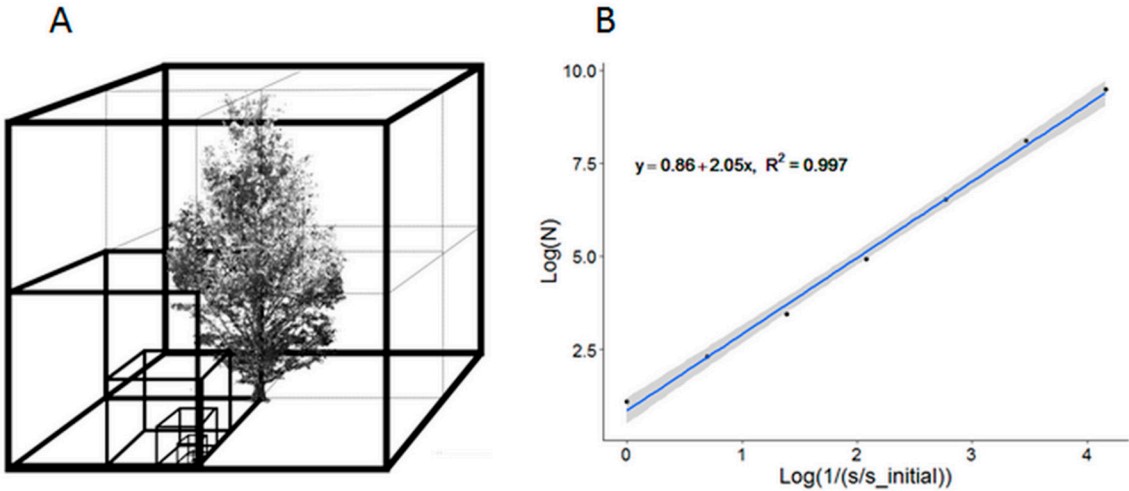

**Figure 2.** (**A**) Illustration of the virtual boxes of different sizes that capture the leaf-on point cloud of a *M. glyptostroboides* tree. (**B**) Exemplary log–log plot for the computation of the box-dimension metric for the same tree. The slope of the regression line equals the box-dimension of the tree, i.e., $D_b$ = 2.05. The 95% confidence interval has been plotted around the regression line. The number of boxes required to capture all points of the tree point cloud is denoted as *N*, the size of the length of each box is denoted as *s*, and the size of the length of the initial box that encapsulates the whole tree is denoted as *s_initial*.

### 2.5. The LCC Difference Index and Error Metric Computation

The role of leaves in the above-ground structural complexity of the trees was quantified with the Leaf Complexity Contribution index

$$\text{LCC} = \left[ \left( \frac{D_b(\text{leaf.on})}{D_b(\text{leaf.off})} \right) - 1 \right] \tag{1}$$

where $D_b(\text{leaf.on})$ is the box-dimension of the leaf-on point cloud of each study tree, and $D_b(\text{leaf.off})$ is the box-dimension of the leaf-off point cloud of each study tree. If LCC = 0, the $D_b$ of the leaf-on and leaf-off point clouds of a tree are equal and there is no contribution of the leaves to the structural complexity of the tree. If LCC > 0, it means that the leaf-on $D_b$ of a tree is greater than the leaf-off $D_b$ of the tree, indicating that leaves increase tree structural complexity. Similarly, if LCC < 0, it means that the leaf-off $D_b$ of a tree is greater than the leaf-on $D_b$ of the tree, indicating that leaves reduce structural complexity, most likely because they occlude the woody components that are not adequately laser-scanned.

The effect of the artificial leaf removal using the *TLSeparation* algorithm on the structural complexity of each study tree was quantified with the percent relative error metric [95,96]

$$\%RE = \frac{|D_b(\text{lf.off}) - D_b(\text{lf.rm})|}{D_b(\text{lf.off})} \times 100 \tag{2}$$

where $D_b(\text{leaf.rm})$ is the $D_b$ of the point cloud of each study tree after the artificial leaf removal.

### 2.6. Computation of Other Structural Metrics of Trees

We computed additional metrics that characterize the structure of trees to test our hypotheses regarding how the LCC index and the %RE relate to the above-ground tree

architecture. According to major theories of tree structural complexity (i.e., pipe model theory [7]; metabolic scaling theory [11]), the "pipes" of the vascular system of a tree connect the roots to the leaves, with a surface area that scales with their volume [97]. Consequently, the structural complexity of the vascular structure of a tree depends on the length and diameter of its pipes [97,98]. Therefore, we expected that the LCC should relate to different metrics of the length of the paths from the base of a tree to each branch tip (e.g., the "path fraction" metric of Smith et al. [99]).

The algorithm *TreeQSM* v.2.3.0 [100] was used to produce quantitative structure models (QSMs) from the leaf-off point clouds of the trees. *TreeQSM* includes two main steps: (1) the point cloud segmentation into stem and branches based on cover sets and (2) the reconstruction of the volume and the surface area of the segments with cylinders [56,101]. The algorithm produced several QSMs for each tree point cloud based on a range of values for the minimum and maximum size of the cover sets and it finally determined the optimal QSM [67]. Based on the parameters of the optimal QSM, the algorithm produced 30 additional QSMs in order to estimate the variation of the modeled tree variables (e.g., woody surface area) because of the inherent stochasticity of the *TreeQSM* algorithm [67]. The algorithm separated the main stem from the branches of a tree based on the following criteria: (1) the main stem extends near the top of a tree, (2) it goes almost straight up, and (3) it is not too curved, which means that the ratio of the stem length to the stem base-tip distance must be the minimum among all candidate main stems; the branches were further categorized by branching order based on certain criteria for branch topology, branch length, and branch base-tip distance (P. Raumonen, personal communication, 2 June 2020).

From the optimal QSMs of the leaf-off point clouds of the study trees, their total woody surface area (the surface area outside of the bark tissues) was computed as the sum total surface area of the cylinders that were fitted to the point cloud of each tree. The total woody surface area of each tree was also separated into the main stem and the branch woody components.

"Path lengths" [99] were also used to create alternative structural metrics of the trees. The lengths of all paths from the stem base of a tree to all branch tips were computed from the lengths of the QSM cylinders, whose topological structure is preserved in a QSM. The distribution of the path lengths for each tree was computed, i.e., the percentiles of the path lengths (25th, 50th, and 75th percentiles), as well as minimum, maximum, and mean path lengths. Smith et al. [99] showed that relative path length variation is an intrinsic element of tree branching architecture relating to tree hydraulic conductance, volume, mechanical stability, and light interception.

### 2.7. Statistical Analyses

All statistical analyses for this study were carried out with custom coding and available packages written in the R software language [102].

Differences in the mean value of the $D_b$ of the trees for leaves-on versus -off, and leaves-artificially removed versus -off, were evaluated with *t*-tests, for each species separately (*G. triacanthos*, *Q. macrocarpa*, and *M. glyptostroboides*), and for all species combined. *T*-tests were also used to evaluate differences in the mean value of the LCC index, %RE, and $D_b$ of leaf-on, leaf-off, and leaf-removed tree point clouds between the study species. The "sma" function of the standardized major axis regression and testing routines ("smatr") R package [103] was used to conduct hypothesis tests regarding the intercepts and the slopes of the species sub-population regression lines. In all statistical tests, significant differences were assessed at $\alpha = 5\%$.

The relationships between the leaf-on, leaf-off, and leaf-artificially removed $D_b$ values, and the relationships of the LCC index and the %RE with the tree structural metrics (see Section 2.6) were analyzed using linear regression analysis and relationship strength was quantified with the Pearson correlation coefficient (*r*); statistical significance was assessed at $\alpha = 5\%$.

## 3. Results

### 3.1. Structural Complexity of Leaf-On versus Leaf-Off Tree Point Clouds

The data show that the study trees varied widely in size (DBH and height) and structural complexity (Table 1). There was significant difference between the mean $D_b$ of the leaf-on tree point clouds of the *G. triacanthos* (GLTR) and *Q. macrocarpa* (QUMA) species ($p$ = 0.0194). However, no significant difference was found between the mean $D_b$ of the leaf-on tree point clouds of the *G. triacanthos* (GLTR) and *M. glyptostroboides* (MEGL) trees ($p$ > 5%), or for the MEGL and QUMA trees ($p$ > 5%). Significant differences were found between the mean $D_b$ values of the leaf-off tree point clouds of QUMA and MEGL trees ($p$ = 0.0335), GLTR and QUMA trees ($p$ < 0.001), and MEGL and GLTR trees ($p$ = 0.041).

*T*-tests showed that the mean $D_b$ of the leaf-on tree point clouds was significantly greater than the mean $D_b$ of the leaf-off tree point clouds (Figure 3) across all study tree species combined ($p$ < 0.001), and for each species separately (GLTR: $p$ = 0.0145; QUMA: $p$ < 0.001; MEGL: $p$ = 0.003). Positive relationships were found between the leaf-on and the leaf-off $D_b$ values of the trees across all species combined (Pearson's $r$ = 0.72, $p$ < 0.001) and for the GLTR (Pearson's $r$ = 0.91, $p$ < 0.001) and QUMA species (Pearson's $r$ = 0.6, $p$ = 0.019) (Figure 4). The relationship between the leaf-on and the leaf-off $D_b$ values for the MEGL trees was not significant (Pearson's $r$ = 0.52, $p$ = 0.055); however, all data points were above the 1:1 line indicating that the $D_b$ of the MEGL leaf-on point clouds was clearly greater than the $D_b$ of the MEGL leaf-off point clouds, except one tree with an LCC index close to zero (LCC = 0.00064) (Figure 4D).

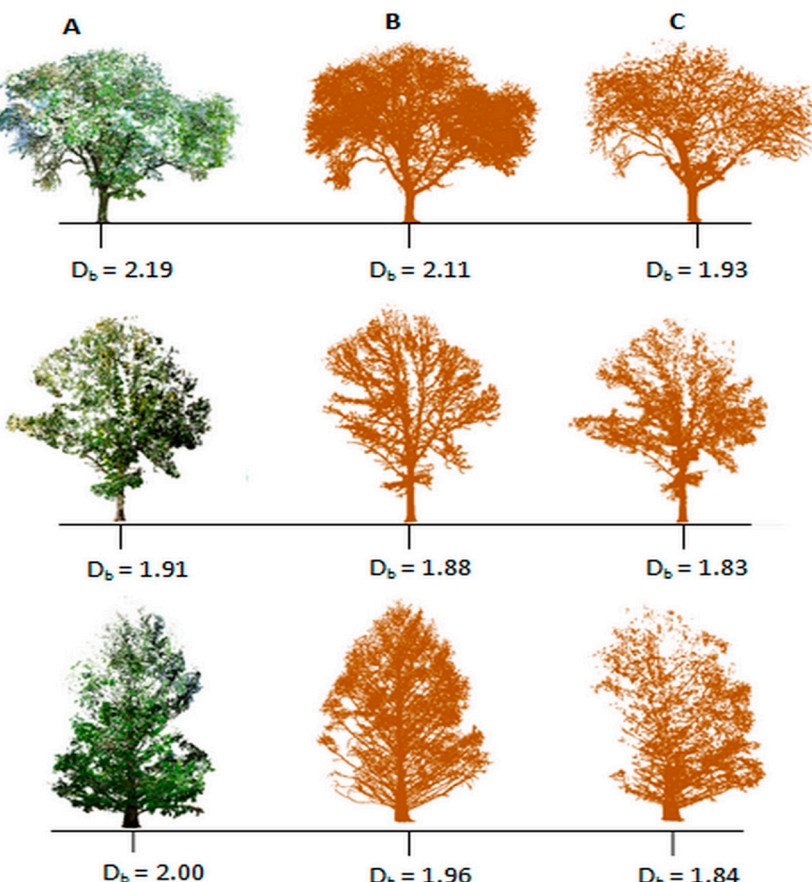

**Figure 3.** Structural complexity quantified with the box-dimension ($D_b$) metric of the (**A**) leaf-on, (**B**) leaf-off, and (**C**) leaf-removed point clouds of a *G. triacanthos* tree (first row), a *Q. macrocarpa* tree (second row), and a *M. glyptostroboides* tree (third row). The leaf-off and leaf-removed tree point clouds have been artificially colored with brown color.

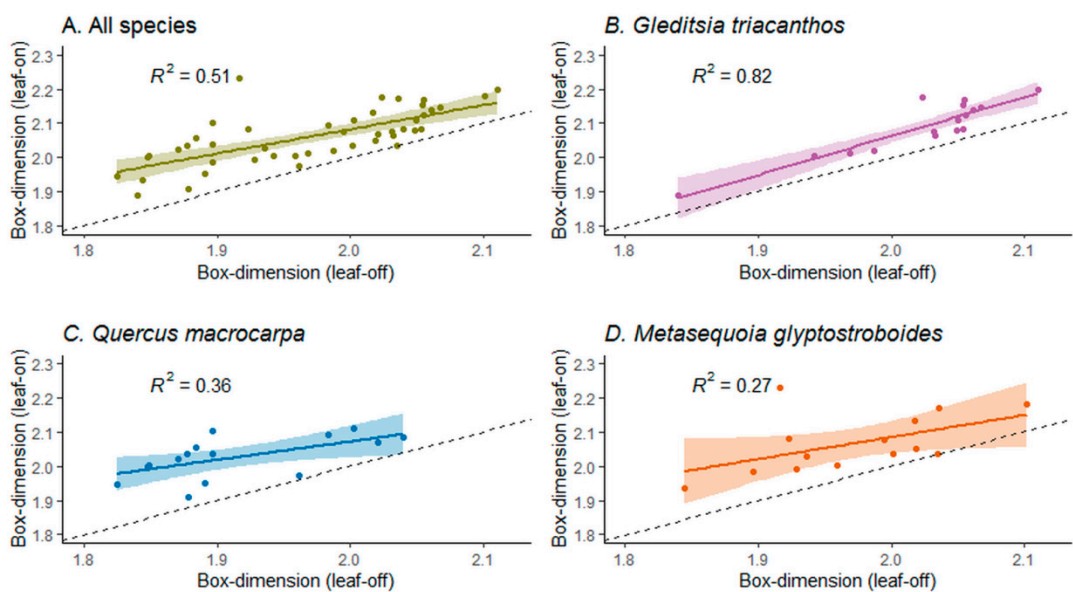

**Figure 4.** Relationship between the leaf-on and leaf-off box-dimension values across all study tree species combined, and for each species separately with 95% confidence interval around the regression lines. The black dashed line is the 1:1 line.

The LCC index ranged between 0.00064 and 0.16394 across all trees combined (see Table 1), indicating a significant reduction in the structural complexity of deciduous tree crowns when leaves are shed. The mean LCC index value was significantly different between GLTR (mean $LCC_{GLTR}$ = 0.03273) and QUMA (mean $LCC_{QUMA}$ = 0.05867) trees ($p$ = 0.0261). However, the mean LCC index value was not significantly different between QUMA and MEGL (mean $LCC_{MEGL}$ = 0.04864) trees ($p$ = 0.4559), or between GLTR and MEGL trees ($p$ = 0.181).

The LCC index was negatively correlated with the branch woody surface area of the study trees (Pearson's $r$ = −0.4, $p$ = 0.0061), but it was not correlated with their stem woody surface area ($p$ = 0.16) (Figure 5). The "outlier" MEGL data point in Figure 5 (point with LCC > 0.15) did not drive the observed relationship because the pattern did not change after the removal of this data point.

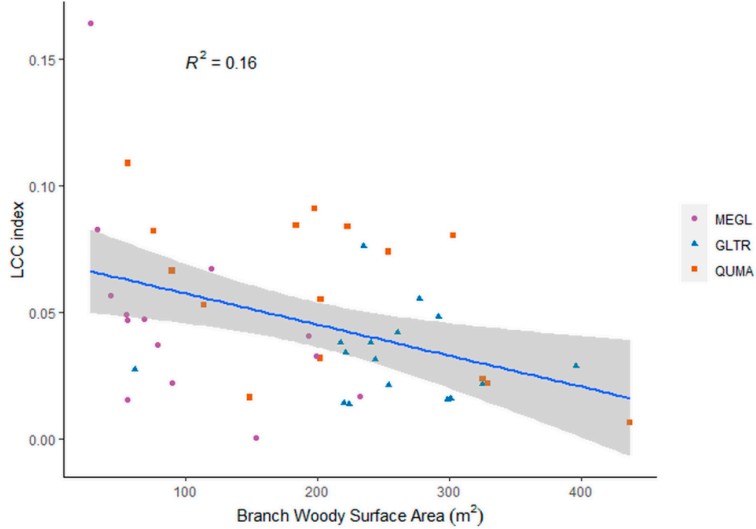

**Figure 5.** Relationship between the LCC index and the branch woody surface area of the trees with 95% confidence interval around the regression line. The three species *M. glyptostroboides* (MEGL), *G. triacanthos* (GLTR), and *Q. macrocarpa* (QUMA) have been plotted with different colors and symbols.

Finally, the LCC index was negatively correlated with different path length variables, i.e., mean path length (Pearson's $r = -0.4$, $p = 0.0068$), maximum path length (Pearson's $r = -0.44$, $p = 0.0025$), and the 25th percentile of path lengths (Pearson's $r = -0.41$, $p = 0.0051$) (Figure 6). The "outlier" MEGL data point in Figure 6 (point with LCC > 0.15 in each graph) did not drive the observed relationships because the patterns did not change after the removal of this data point.

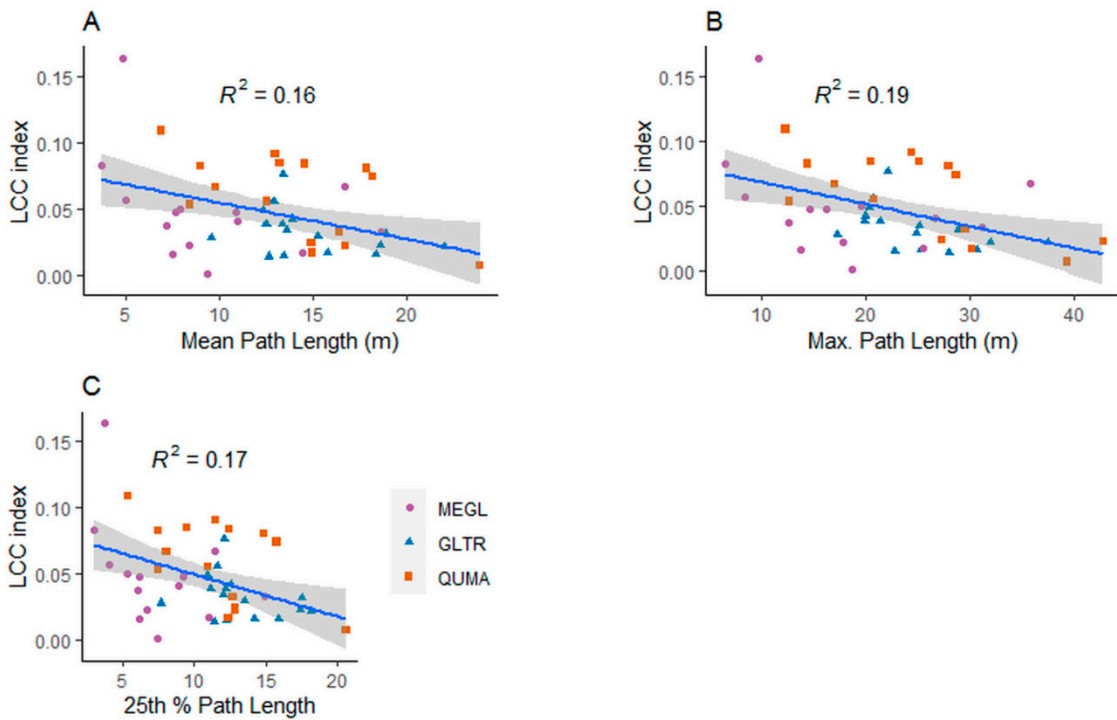

**Figure 6.** Relationships between the LCC index and different path length variables with 95% confidence interval around the regression lines. The three species *M. glyptostroboides* (MEGL), *G. triacanthos* (GLTR), and *Q. macrocarpa* (QUMA) have been plotted with different colors and symbols.

### 3.2. Box-Dimension of Leaf-Off versus Leaf-Removed Tree Point Clouds

Significant differences were found between the mean $D_b$ values of the tree point clouds after the artificial leaf removal for QUMA and GLTR trees ($p = 0.001$), and GLTR and MEGL trees ($p = 0.0105$), but no significant difference was found between the mean $D_b$ of the MEGL and QUMA trees after the artificial leaf removal ($p = 0.9662$).

*T*-tests showed that the mean $D_b$ of the leaf-off tree point clouds was significantly greater than the mean $D_b$ of the leaf-removed tree point clouds across all study tree species combined ($p < 0.001$), and for the GLTR trees ($p < 0.001$). No significant difference was found between the mean $D_b$ of the leaf-off and leaf-removed point clouds for the QUMA trees ($p = 0.6382$), or the MEGL trees ($p = 0.1622$). Furthermore, the leaf-removed and the leaf-off $D_b$ values of the QUMA trees were positively correlated (Pearson's $r = 0.65$, $p = 0.0082$), but no significant relationship was found between the leaf-removed and the leaf-off $D_b$ values across all study tree species combined ($p > 5\%$), or for the GLTR and MEGL trees ($p > 5\%$) (Figure 7). The standardized major axis tests showed that the intercept and the slope of the regression line of the QUMA trees was not statistically different from the 0 and 1 values, respectively.

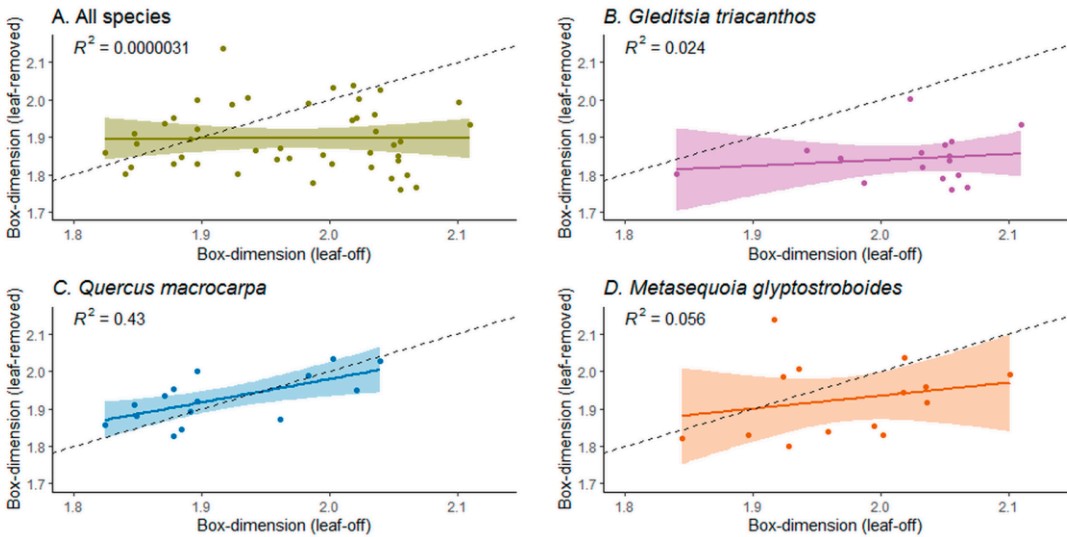

**Figure 7.** Relationship between the leaf-removed and leaf-off box-dimension values across all study tree species combined, and for each species separately with 95% confidence interval around the regression lines. The black dashed line is the 1:1 line.

The mean $D_b$ of the leaf-on tree point clouds was significantly greater than the mean $D_b$ of the leaf-removed tree point clouds (Figure 3) across all study tree species combined ($p < 0.001$), and for each species separately (GLTR, QUMA, MEGL: $p < 0.001$).

The mean %RE value was significantly different between GLTR (mean %RE$_{GLTR}$ = 8.91%) and MEGL (mean %RE$_{MEGL}$ = 5.06%) trees ($p = 0.0057$), and between GLTR and QUMA (mean %RE$_{QUMA}$ = 2.43%) trees ($p < 0.001$), and also between MEGL and QUMA trees ($p = 0.0064$).

The %RE was positively correlated with the maximum branch order of the GLTR trees (Pearson's $r = 0.53$, $p = 0.033$), but it was not correlated with the maximum branch order of the QUMA and MEGL trees ($p > 5\%$) (Figure 8).

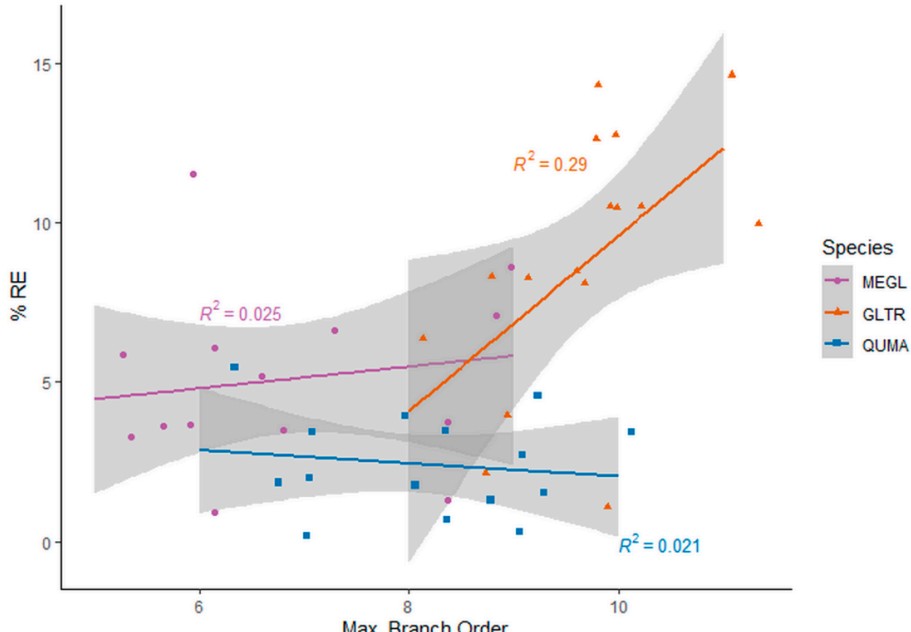

**Figure 8.** Relationship between the % Relative Error (RE) and the maximum branch order of the trees with 95% confidence interval around the regression lines. The species *M. glyptostroboides* (MEGL), *G. triacanthos* (GLTR), and *Q. macrocarpa* (QUMA) have been plotted with different colors and symbols.

## 4. Discussion

### 4.1. Structural Complexity of Urban Trees

This study measured the $D_b$ of the above-ground components of tree architecture (i.e., main stem, branching network, and leaves) from TLS point clouds, to determine the above-ground structural complexity of trees growing in urban areas. $D_b$ can help to understand how trees maximize resources uptake for their growth while maintaining their mechanical stability [13,90,93]. From an evolutionary perspective, trees have had to develop an "adaptive" geometry [104] to optimize light capture and minimize self-shading [18,91,92], while balancing with other competing functions, such as maintaining mechanical stability [77] and resisting drought [36]. Open-grown trees are relatively free from light competition due to having fewer tree neighbors [77], so they are more likely to be able to maximize their structural complexity and express their inherent fractal-like architecture than trees growing in forests or plantations [17]. The urban open-grown trees in this study were not directly influenced by shading from neighboring trees or from the relatively short buildings that were near to some of the trees. $D_b$ is sensitive to the external shape and the internal structure of trees [90,93], so differences in $D_b$ can capture meaningful differences in tree architecture and physiological function. Therefore, it is important to consider what the maximum structural complexity could be.

Seidel et al. [90] hypothesized that trees should have $D_b$ values significantly lower than 2.72, which is the $D_b$ of the Menger sponge (a mathematical object with the greatest surface to volume ratio, [105]), assuming a tree would maximize its surface area for light capture and gas exchange, while minimizing building costs, in the absence of competition with other plants. In previous studies that quantified the above-ground complexity of trees growing in dense rural forest stands, leaf-on $D_b$ values were consistently lower than 2 [13,19,89,90,93,106]. In this study, the mean $D_b$ of the leaf-on tree point clouds was greater than 2 across all study tree species (see Table 1), indicating a possible structural difference between trees in rural versus urban areas. However, rural forest trees growing in more open conditions and facing less competition for light (e.g., in forest gaps and in thinned forest stands), also had larger $D_b$ values [13,19,106], in some cases exceeding 2 [107]. This suggests a benefit to having an increased $D_b$ with more light and fewer neighbors, but at some level the energy benefits from increased photosynthesis would be minimized due to a high level of self-shading [90]. This supports MacFarlane et al.'s [17] assumption that trees growing in the open, without competition, can more closely approach the theoretical maximum $D_b$ (as characterized in Seidel et al. [90]). In this study, the maximum $D_b$ value observed was 2.23, for a large specimen of *M. glyptostroboides* in the leaf-on condition (Table 1). So, even the largest, open-grown, urban trees in this study were well below the theoretical maximum of 2.72.

### 4.2. The Role of Leaves in the Structural Complexity of Deciduous Trees

The urban trees studied here were deciduous species, characterized by distinct leaf phenological changes (i.e., leaf-on and leaf-off periods), which are typically affected by sharp photoperiodic and temperature changes [108,109]. In general, the outline shape and the texture of leaves can have fractal-like patterns [86–88,110–116], and thus, we expected that the presence of leaves would increase the total structural complexity of trees. Indeed, the study trees were shown to have statistically different structural complexity in the leaf-on and leaf-off periods (Figure 4) because the presence of leaves implies greater dispersion of laser points in the leaf-on point clouds compared to the leaf-off point clouds and more boxes are required to capture them, which results in greater value for the $D_b$ metric and greater structural complexity [13,89,90]. In a previous study, the difference between the $D_b$ of the leaf-on and leaf-off point clouds of forest-grown trees was not significant [89]. However, that study followed a mixed approach to generate leaf-off point clouds. More specifically, from the 76 leaf-off point clouds, only 15 point clouds were captured during the leaf-off period and the remaining leaf-off point clouds were created after manual segmentation of leaves from the leaf-on point clouds [89].

The magnitude of change in $D_b$ observed in this study was relatively small; the LCC index ranged from 0.00064 to 0.16394 across all species combined, indicating that the largest portion of the total above-ground structural complexity of a tree comes from woody components, e.g., branches. However, $D_b$ is constrained to have values between one and three, so a small change in its value can have significant physiological implications. Seidel et al. [90] found that the crown surface area divided by the woody volume of trees increased as a power function of leaf-on $D_b$, so that, for example, an increase of 0.2 units in leaf-on $D_b$ resulted in approximately 40 units of increase in crown surface area relative to the woody volume of trees. Similarly, the results here in this study show that a small change in the structural complexity has important implications for urban trees. An increase of approximately 0.05 units in the LCC index was associated with approximately 400 m$^2$ reduction in the branch woody surface area of the study trees (Figure 5). Such a change could have important implications for the mechanical stability of trees, i.e., the branch woody surface area affects the bending moments due to wind drag [30,79,117], for the maintenance respiration of trees that relates to their woody surface area [118–122], and for solar radiation and rainfall interception [123].

Differences in the LCC index were related to other structural metrics of the trees, showing different changes in the $D_b$ value, with and without leaves, for different types of trees. The negative relationships between the LCC index and the branch surface area and the path length metrics indicate that larger trees, with larger and more "branchy" crowns, have a relatively smaller contribution of leaves to structural complexity (Figures 5 and 6). These results can be interpreted within the framework of the pipe model theory [7] and the West–Brown–Enquist or WBE model [11,124], which explain the fractal-like architecture of trees by assuming a vascular tree structure consisting of pipes [11]. According to these theories, as the size (i.e., woody surface area or length) of the pipes of the vascular system of a tree increases, the structural complexity of the woody skeleton of the tree also increases.

Differences in species branching architecture and leaf structure could also explain some of the observed differences in leaf-on versus leaf-off $D_b$ values because the fractal architecture of urban tree crowns is influenced by both crown and leaf shape [36]. *G. triacanthos* trees had the smallest contribution of the leaves to the structural complexity (smallest LCC). According to Niinemets and Valladares [125], *G. triacanthos* is the least shade tolerant of the three species studied (shade tolerance index for *G. triacanthos* = 1.61, *Q. macrocarpa* = 2.71, and *M. glyptostroboides* = 3). Species which are very shade tolerant distribute their leaves more evenly within their crown volume [36], whereas species that are less shade tolerant, e.g., *G. triacanthos*, have their leaves widely spaced mainly in the crown periphery, in order to increase crown porosity and reduce local self-shading [91]. Furthermore, it has been suggested that inter-canopy variation of leaf traits is predominantly affected by the exposure of leaves to light, which makes the sun leaves that are distributed in the crown periphery smaller, with greater leaf mass per unit area compared to the crown-interior leaves, in order to reduce water loss through transpiration [91,92]. Therefore, the uneven distribution of leaves in the crown volume of the *G. triacanthos* trees, most of which are small sun leaves in the crown top, could explain why the contribution of leaves in the overall structural complexity was the smallest when compared to *Q. macrocarpa* and *M. glyptostroboides* trees.

### 4.3. The Effect of the Leaf Separation Algorithm on the Structural Complexity of the Trees

Very often, one is unable to laser-scan trees during the leaf-off conditions, either because they are evergreen or due to logistical constraints. Therefore, one of the goals of this study was to explore the effect of artificial leaf removal from leaf-on point clouds. Separating the woody component from the foliage of tree point clouds using classification algorithms is a challenging task. Zhu et al. [126], for example, found a significant overestimation in the leaf area index of trees because of the woody material in tree point clouds.

There are different algorithms and approaches to separate leaves from the woody structure of tree point clouds [80–83], but there is no single best solution that fits for all point

classification cases in forests [80]. Some of the factors that influence the classification results are the following: heterogeneity of point cloud density, varying scanner configurations, and scanning protocols [80]. The *TLSeparation* algorithm, which was used here, does not depend on a specific scanner [81], and we tried to minimize the occlusion effects in the point clouds in this study by scanning each tree from multiple directions and distances at high scanning resolution, following the field scanning protocols suggested by Wilkes et al. [69]. The performance of leaf separation algorithms is significantly decreased by occlusion [81], but explicit accounting of this error source is challenging because we don't have complete control over it, and different types of error can be correlated [95].

Errors in characterization of crown architecture should relate to leaf morphology [80]. Wang et al. [83] suggest that leaves are typically detected as simple, flat structures and, therefore, the oblong leaf shape or the modular structure of compound leaves might confuse the classification algorithms. Indeed, our results indicate that the *TLSeparation* algorithm can be more accurate in identifying simple flat leaves, but had more difficulty separating twigs and fine branches from compound leaves. The *Q. macrocarpa* trees showed no statistical difference in mean $D_b$ of the leaf-off and leaf-removed point clouds and this species has simple leaves with a single flat and lobbed blade (or lamina) [127], which is associated with important leaf physiological functions, e.g., convection heat dissipation, efficient light interception, and reduced leaf hydraulic resistance [91,111]. The *TLSeparation* algorithm [85] appears to have miss-classified many points of the woody structure as leaves for the *G. triacanthos* trees, which have compound leaves with a modular architecture, because the leaf blade consists of several leaflets stemming from the leaf rachis [128,129]. The *TLSeparation* algorithm added significant noise into characterizations of $D_b$ in *M. glyptostroboides* trees, which are deciduous gymnosperms and have oblong-shaped needles and branches that are either horizontal or curved upward [130]. We might expect the accuracy of the *TLSeparation* algorithm for needle-leaved trees to be lower compared with the classification accuracy of broad-leaved trees because needles are linear and it is difficult to resolve an individual needle due to its small size and the dense foliage of conifers [81,83]. In a previous study, it was found that artificial leaf removal using a different leaf separation algorithm (i.e., *LeWoS* algorithm) resulted in the underestimation of the total woody volume of trees in the generated QSMs, while only the stems and some large branches were detected in coniferous trees [83].

As was originally hypothesized, the percent relative error in the estimated structural complexity of the *G. triacanthos* trees, after artificial leaf removal, was related to the branching architecture of the trees. More specifically, trees of this species with higher maximum branch order had greater %RE values, indicating that the presence of more bifurcations (branching nodes) and smaller branches of higher order can reduce the accuracy of the *TLSeparation* algorithm to classify the leaves and the woody parts. Indeed, increased branch bifurcation and angulation result in increased occlusion in the point clouds of trees that reduces the accuracy of the leaf-classification algorithm [82]; in a previous study the point density of woody structures decreased for higher branch orders and, therefore, many points were miss-classified as leaves [83]. The %RE values of the *M. glyptostroboides* and *Q. macrocarpa* trees were not related to their maximum branch order, presumably because the leaf removal algorithm did not significantly affect the accuracy of the $D_b$ of the *Q. macrocarpa* and *M. glyptostroboides* trees on average according to the *t*-tests, although the $D_b$ of the *M. glyptostroboides* trees after the artificial leaf removal was imprecise.

## 5. Conclusions

This study used terrestrial laser scanning (TLS) to further refine our understanding of the above-ground structural complexity of urban trees by separating the effect of leaves from the effect of the woody skeleton. Differences in leaf-on versus leaf-off structural complexity likely relate to different functional traits of trees for light capture optimization, reduction of self-shading, and mechanical stability. As such, this study provides evidence that differences in the contribution of leaves to tree structural complexity could be an

important indicator of where the plant lies on a "structural economics spectrum (SES)", which explains species structural diversity in terms of tree architectural traits along a spectrum balancing light interception, carbon allocation, and mechanical stability [131]. However, more species belonging to different functional groups must be included in future studies in order to further examine differences in the LCC, or a similar index, as part of the SES. This study provided evidence, along with previous studies [80,83], that the accuracy of leaf separation algorithms is affected by the leaf shape and type, but also that bias in the estimation of the above-ground structural complexity of trees after the artificial leaf removal depends on the branching architecture.

**Supplementary Materials:** Available online at https://www.mdpi.com/article/10.3390/rs1314277 3/s1 can be found the aim and purpose, applicability, theory and background, and the full code of the box-dimension algorithm for one or multiple data files.

**Author Contributions:** G.A. and D.W.M. conceived the ideas and designed the methodology for the study; G.A., D.W.M. and D.S. analyzed the data; G.A., D.W.M. and D.S. led the writing of the manuscript. D.S. wrote the description of the algorithm that computes tree box dimension. All authors have read and agreed to the published version of the manuscript.

**Funding:** This work was partially supported with funds from a joint venture agreement between Michigan State University and the United States Department of Agriculture Forest Service, Forest Inventory and Analysis Program, Northern Research Station. Part of D.W. MacFarlane's time was paid for with funds from Michigan AgBioResearch, the USDA National Institute of Food and Agriculture. Part of G. Arseniou's time was supported by a Bouyoukos Fellowship. Part of D. Seidel's time was supported by the German Research Foundation (DFG) through the grant SE2383/7-1.

**Data Availability Statement:** The data presented in this study are available on request from the corresponding author.

**Acknowledgments:** We want to acknowledge the Michigan State University Campus Arboretum (Frank W. Telewski, and Jeffrey Wilson) and the Michigan State University Department of Infrastructure, Planning and Facilities (Jerry Wahl) who permitted and assisted with the data collection.

**Conflicts of Interest:** The authors declare no conflict of interest.

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
