# Peer review of "Measuring the Contribution of Leaves to the Structural Complexity of Urban Tree Crowns with Terrestrial Laser Scanning"

_remotesensing, doi:10.3390/rs13142773_

Round 1

Reviewer 1 Report

Dear Authors,

Thank you very much for this interesting and current contribution. The manuscript „Measuring the Contribution of Leaves to the Structural Complexity of Urban Tree Crowns with Terrestrial Laser Scanning” researches the effect of leaves and artificial leave removal to structural metrics like the box dimension of urban trees. The manuscript is in an advanced state and well written. The contend is presented in a clear and not overly complicated way, the figures are clear and the captions helpful.

The introduction is partly a little bit wordy, especially the part tackling ecosystem services (L. 65 …).

The authors show an impressive amount of literature in the introduction, even for the points which are of less importance for the manuscript. This is indeed not wrong but I would prefer to be a little more selective, to make it easier to find the content between the citations.  

Your research questions seem to be relevant to me, despite the second one (L. 153). I understand why you want to know what influence it has if trees are scanned leaves on/off, but I don’t get the message the LCC should deliver in an ecological sense. Therefore, I don’t understand why you want to relate it to structural complexity measures. I would cut out the whole question and all methods, results and discussions part related to it, but I am open to your explanations and arguments.  

The methods are clearly described. I would prefer if you would include the abbreviation to the tree species at their first occurrence in the manuscript (L. 187 / 327) and include their English names.

I was a little confused, that the box dimension was explained partly in the introduction and partly in the methods section at first reading, but I don’t have a good solution for this problem since the technical part wouldn’t really fit the introduction.

I am not familiar with the smart package (L 312) . Could you add a sentence which functions you used for which purpose?

According to your results, could you please explain to me how you achieved the result ’Significant differences were found among the mean Db values of the leaf-off tree point-clouds across all three species 331 combined (p < 0.001)’ (L 330). I was not aware such thing as a three-sided t-test exists.

According to your table 1 the leaf-off pcs of the tree species have very similar mean values and strongly overlapping data ranges, which makes me wonder how these can be significantly different. Could you maybe provide box-plots of these species comparisons?

In figure 3 and all other scatterplots with linear regressions, please use homogeneous axis scaling between x and y axis and between plots within one figure if they are meant for comparison.

For a couple of plots the confidence band is quite narrow compared to the wide range of the data points, please dubble check that these are the right confidence bands.

Please be consistent between your plots and the text according to the slope abbreviation (r/R). I don’t think r/R is the right abbreviation for the slope. I would write y = <slope>*x + <intercept> within the plot, if I would like to report these values.

Please report the R² values for your regressions.

‘Therefore, the LCC index that quantifies the contribution of leaves to the structural complexity of trees could be an important metric of the plant "structural economics spectrum", which explains species-structural diversity in terms of tree architectural traits along a spectrum balancing light interception, carbon allocation and mechanical stability’ (L 596). I don’t think the results you showed justify your conclusion, since neither the scatterplots nor the values given in table 1 show that the LCC shows real differences between the tree species which obviously have different structural economics spectra. Please rethink this conclusion or justify it.

I hope these comments are helpful to you to improve your manuscript.

Best regards

Author Response

Dear reviewer,

Thank you for your valuable feedback based on which we made important changes to our paper in order to provide clarity. We appreciate your time and your comments. Because the platform allows to upload only one document, I have combined in one single word file our point-by-point responses to your comments and the revised manuscript. Please see the attachment.

Reviewer 2 Report

Dear Authors,

The present study describes the use of TLS data to enhance our understanding of canopy complexities. I believe the presented experiment and methodology are a very valuable addition to knowledge in this field.

I have a few minor suggestions to improve your manuscript.

Introduction

  • This section is hard to understand. I would suggest you rewrite it (L179-184)
    “Finally, we hypothesized that errors in estimation of the box-dimension resulting from artificial leaf removal, would relate to the type of leaf (broad vs. needle and compound vs. simple) and the order and the size of the branches of a tree, the latter of which due to the sensitivity of the leaf separation algorithms to point cloud density, which can change across the branching network of a tree (Vicari et al. 2019, Moorthy et al. 2020). “

Methods

  • (Section 2.1.) I suggest you add pictures of each tree species and leaf unit. This would help readers that are not familiar with these species to visualise the differences.

  • (Section 2.1.) “..pruning did not cause any bias in…” (L 198). Were these trees pruned prior to the study? This would influence the tree architecture even if it was done before.

Results

  • If you cannot add pictures of each tree, I would suggest adding the other 2 species on Figure 2.

Discussion

  • (Section 4.4) In L 525, you stated “Very often, one is unable to laser-scan trees during the leaf-off conditions (…)”. It would be nice to have a brief explanation of why leaf-off scanners are important.

Conclusion

  • Just a general comment about urban trees. Not all urban trees will be growing free of constraints (e.g. street trees) and even if they do not have other trees around, there will be other structures competing with canopy and roots systems, which can influence in expression of the tree architecture.

Author Response

(The authors gave the same response as above.)

Round 2

Reviewer 1 Report

Dear Authors,

thank you very much for the improvements. I think the figures strongly increased in quality and the boxplot convinced me, that the Db actually shows a strong signal to the species. According to the LCC as an indicator for the functinal traits of tree species, I am still not fully convinced, but I guess this is partly a matter of taste and I eave it to the readers to judge this.

I wish you all the best with your publication.